# APPLICATION OF GAUGE EQUIVARIANT CONVOLUTIONAL NEURAL NETWORKS TO LEARNING A FIXED POINT ACTION FOR SU(3) GAUGE THEORY

**Kieran Holland**
University of the Pacific
3601 Pacific Ave., Stockton, CA 95211, USA
kholland@pacific.edu

**Andreas Ipp & David I. Müller**
Institute for Theoretical Physics, TU Wien
Wiedner Hauptstraße 8-10/136, A-1040 Vienna, Austria
{ipp,dmueller}@hep.itp.tuwien.ac.at

**Urs Wenger**
Albert Einstein Center for Fundamental Physics
Institute for Theoretical Physics, University of Bern
Sidlerstraße 5, 3012 Bern, Switzerland
wenger@itp.unibe.ch

## ABSTRACT

Lattice gauge theory is pivotal in understanding nuclear physics and the strong interaction of quarks and gluons from first principles, shedding light on phenomena such as confinement and asymptotic freedom, and providing quantitative understanding of masses and decay rates of mesons and baryons. Scaling up corresponding Monte Carlo simulations faces challenges such as critical slowing down and topological freezing. One proposed approach to address these challenges is through the use of fixed point lattice actions. These actions preserve continuum classical properties even after discretization, thereby reducing lattice artifacts at the quantum level, but they can only be defined implicitly. Here, we employ machine learning, specifically lattice gauge equivariant convolutional neural networks (L-CNNs), to learn fixed point actions in a gauge symmetry preserving way. We obtain a fixed point action for four-dimensional SU(3) gauge theory which is superior to previous hand-crafted parametrizations. This advancement is crucial for future Monte Carlo simulations.

## 1 INTRODUCTION

Lattice regularization is the tool of choice to study nonperturbative properties of quantum field theories starting from first principles (Wilson, 1974). Modern lattice QCD simulations have attained a high level of precision and for some important Standard Model quantities, e.g., the QCD coupling at the electroweak scale $\alpha_S(\mu = m_Z)$, they provide the current most accurate determination (Aoki et al., 2022). Increased precision has amplified systematic issues relevant to any lattice calculation, such as the extrapolation to the continuum limit. Numerical simulations become rapidly more costly as the lattice spacing is reduced to zero, not only due to the increased resolution at fixed physical volume, but also due to the increased autocorrelation times (*critical slowing down*) in generating statistically independent samples in Monte Carlo (MC) Markov chains and the related problem of suppressed tunneling between sectors of different topological charge (*topological freezing*) (Schaefer et al., 2011). For a robust continuum prediction, a range of lattice spacings is necessary, requiring a delicate balance between the control of discretization artifacts on coarse lattices on the one hand and the increased cost of simulating on finer lattices on the other.

Here we propose to use a so-called fixed point (FP) action to solve both critical slowing down and topological freezing. Such an action in principle allows simulations on very coarse lattices where both problems are absent, while at the same time lattice artifacts can be kept so small that a solid continuum limit can be taken. FP actions can be implicitly defined via renormalization group transformations and they can completely remove lattice artifacts at the classical level (Hasenfratz

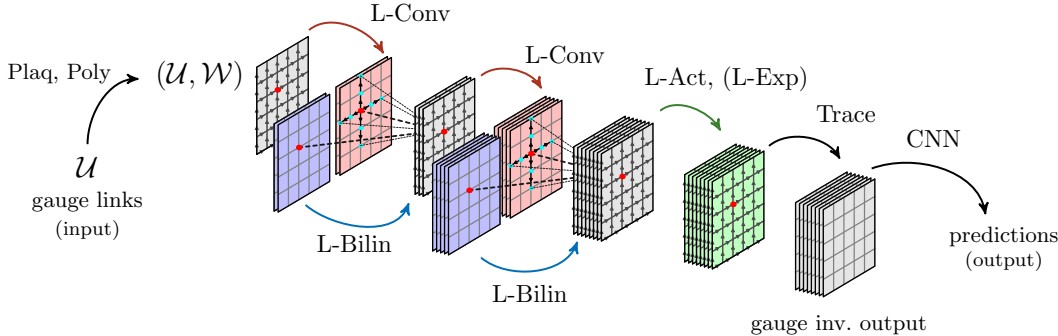

Figure 1: An example of a lattice gauge equivariant convolutional neural network (L-CNN), taken from Favoni et al. (2022). Given a lattice gauge configuration as input, a sequence of layers builds untraced loops of gauge links of increasing size, with the total number of loops increasing rapidly with the depth of the network. The loops are traced in the final layer to produce gauge invariant output. Exact gauge covariance is maintained throughout.

& Niedermayer, 1994). Since they are expected to exhibit suppressed lattice artifacts also at the quantum level, this approach has been used in various MC simulations (DeGrand et al., 1995b;a; 1996; Bietenholz & Wiese, 1996; Blatter et al., 1996; Blatter & Niedermayer, 1996; Niedermayer et al., 2001; Gattringer et al., 2004; Hasenfratz et al., 2004; 2005; 2009). However, practical implementation remains challenging as many FP properties are only implicitly defined, and the FP action requires infinitely many loop operators to describe gauge link couplings. Effective renormalization group transformations (RGTs) have been designed to optimize this process, as demonstrated in previous studies Blatter & Niedermayer (1996). Details on the theoretical setup of the FP action and on our data generation are presented in Appendix A. One is then still left with the challenging task of finding a compact and accurate parametrization of the FP lattice action.

Recent advances in machine learning (ML), in particular the construction of lattice gauge equivariant convolutional neural networks (L-CNNs) by Favoni et al. (2022), now provide a completely new way to tackle this problem. Rather than committing to a particular ansatz for the lattice action, e.g., in terms of some of the smaller closed loops like the plaquette and rectangle, one can have a much more general and expressive neural network architecture, where an optimal set of parameters can be found using ML techniques once a sufficiently rich training dataset is provided. An essential element is that gauge symmetry must be exactly preserved in the network architecture. This has been achieved by starting with the original gauge links and local untraced plaquettes, and creating extended closed loops of gauge links through successive layers using parallel transport and bilinear products of local gauge equivariant operators. In this way, a rapidly increasing number of possible loops is generated with each additional layer. This was shown to be far superior to convolutional neural networks (CNNs) where gauge symmetry was not built into the architecture. The complete generality of the L-CNN approach makes it an ideal method to parametrize FP actions. More details on L-CNNs and the corresponding gauge equivariant neural network model can be found in Appendix B.

## 2 GAUGE EQUIVARIANT NEURAL NETWORK MODEL

L-CNNs built from multiple bilinear convolutions with a final trace layer at the end of the network can be used to express a large class of gauge invariant scalar functions on the lattice. As depicted in Fig. 1, the full architecture can have alternating convolutional and bilinear layers (or combinations thereof), building up more and more loops of increasing length. In principle, any arbitrary loop can be generated once sufficiently many layers are combined. The model also has the possibility to add activation and exponentiation of the variables $W_{x,a}$, which are not used in this particular work. As a final layer, a trace over the variables produces a gauge invariant scalar. Favoni et al. (2022) used L-CNN models to accurately predict traces of planar Wilson loops of size up to $4 \times 4$ in SU(2) gauge theory and found them to be far superior to CNN models which were not constructed with exact gauge invariance.

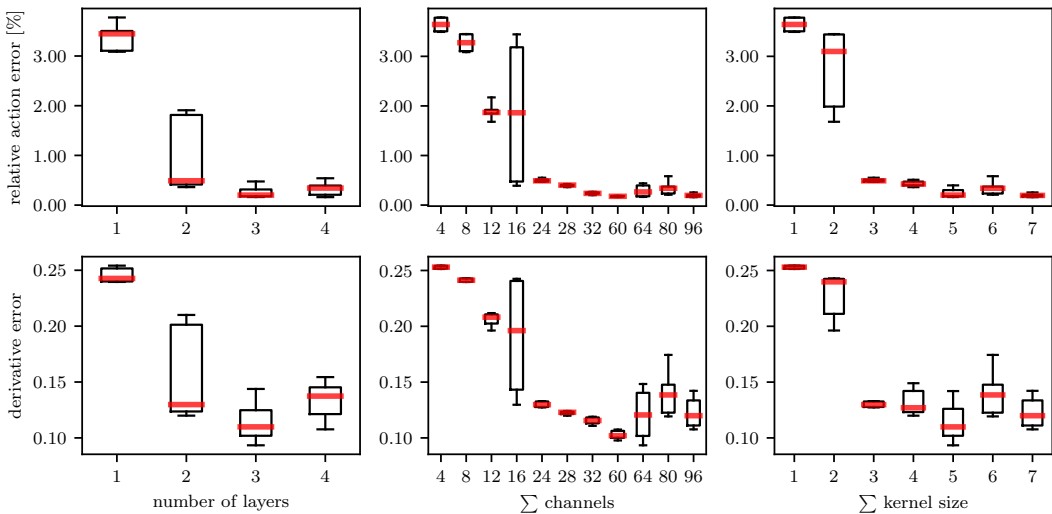

Figure 2: Results of training an ensemble of 130 models, ranging from small to large architectures, on lattice volume of size $4^4$ with $\beta_{\text{wil}}$ ranging from 5.0 to 10.0. Test data consisted of the same lattice size and $\beta_{\text{wil}}$ range. All models use the Wilson action as a prefactor action. We show box plots of the relative errors and derivative errors averaged over all test data. The thick central lines show the median error. The box extends from the 25% to the 75% quantile and the whiskers denote the 0% (minimum) and 95% quantile (to remove outliers). The left panels show the dependence of the errors on the model depth, i.e., the number of bilinear convolutional layers. The middle panels show the dependence on the model width given by the sum of channels across all layers. The right panels show the dependence on the size of the receptive field of the models, which we approximate by the sum of kernel sizes in each layer. We observe that larger models (more layers, more channels, larger receptive field) typically lead to better approximations of the data.

However, there are a few additional requirements to parametrize gauge invariant actions. The first requirement is a normalization condition: the output of a parametrization $\mathcal{A}^{\text{L-CNN}}[V]$ must approach the Yang-Mills action $S_{\text{YM}}[A_\mu(x)]/\beta$ for sufficiently smooth gauge configurations $V_{x,\mu} \approx \exp(iaA_\mu(x))$ with gauge fields $A_\mu(x)$. Secondly, one may require that the naive continuum limit for lattice spacing $a \to 0$ is reached in a particular way such that lattice artifacts of certain observables are suppressed to some desired order, along the lines of Symanzik improvement. A third condition is that the parametrized action should be positive for all gauge configurations. Finally, we require the action to be local, which means that the parametrization should be expressible as a sum over lattice sites of finite-length Wilson loops and their products. All four requirements can be explicitly realized by choosing a particular ansatz for the parametrization model:

$$\mathcal{A}^{\text{L-CNN}}[V] = \sum_x \mathcal{A}_x^{\text{pre}}[V] \sum_{n=0}^{\infty} b^{(n)} (N_x[V] - N_x[\mathbb{1}])^n, \tag{1}$$

where $\mathcal{A}_x^{\text{pre}}[V]$ is the local contribution of a *prefactor action*, $\mathcal{A}^{\text{pre}}[V] = \sum_x \mathcal{A}_x^{\text{pre}}[V]$. The term $N_x[V]$ is the local output of an L-CNN and $N_x[\mathbb{1}]$ is the network evaluated on a link configuration of unit matrices. Finally, $b^{(n)}$ are manually chosen coefficients with the constraint $b^{(0)} = 1$. As we will show below, the prefactor part is used to control the naive continuum behavior of the action, whereas the L-CNN provides corrections for coarse configurations. For details about these requirements and the shape of the prefactor as well as the choice of the loss function, we refer to Appendix B

## 3 RESULTS

We show a summary of the hyperparameter scan in Fig. 2, with 130 L-CNN models used to estimate the distributions, examining the accuracy in predicting the FP action value and the FP derivatives. To compare a variety of models, we study their performance in terms of the model *depth*, the model

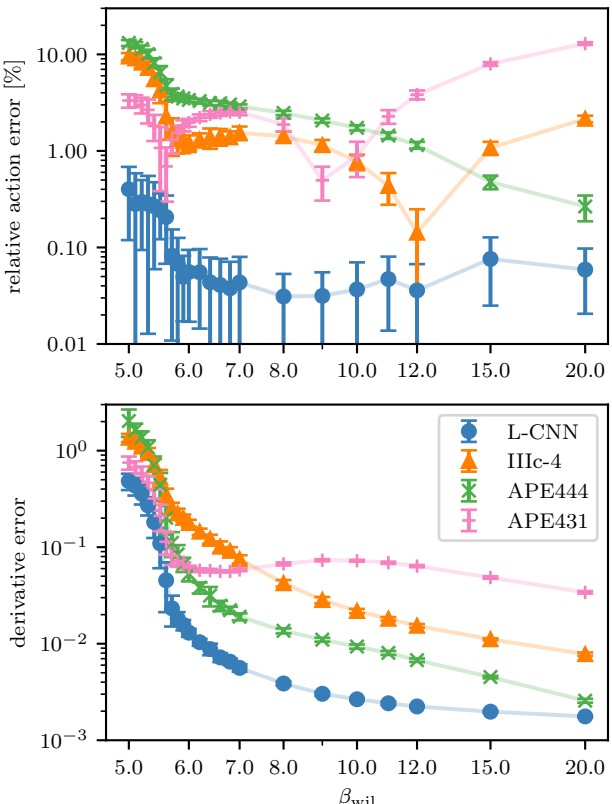

Figure 3: Comparison of different parametrizations of the FP action evaluated on MC ensembles with values of the bare coupling ranging from $\beta_{\mathrm{wil}} = 5.0$ to $\beta_{\mathrm{wil}} = 20.0$ on $4^4$ lattice volumes. The errors of a particular parametrization are defined as the deviations from the numerical fixed point data. The top panel shows the relative error $\mathcal{L}_1$ computed from action values. The bottom panel shows the gauge invariant derivative error $\mathcal{L}_2$. The error bars are given by the standard deviation within each ensemble. Our best machine learned model (L-CNN, blue line) has the smallest error over the whole depicted range of bare couplings, improving results of previous hand-crafted parametrizations denoted as IIIc-4, APE444, and APE431.

*width*, and the size of the *receptive field*. The depth of the model is determined by the number of layers, while the width is related to the number of channels in each layer. As a simple measure of the model width, we take the sum of the number of channels in each layer. The size of the receptive field, which limits the locality of the action, is approximated by the sum of the kernel size for each layer. A general trend is clear: Increasing the depth, width, or receptive field reduces both the action and derivative errors, as one might expect.

In Fig. 3 we compare the older FP parametrizations with the best L-CNN model on gauge ensembles with the bare coupling $\beta_{\mathrm{wil}}$ varied in a range from 5.0 to 20.0. We see that the L-CNN clearly outperforms the previous parametrizations across this range, with its predicted action value and derivatives much closer to the ground truth FP values. Even on much smoother gauge ensembles at $\beta_{\mathrm{wil}} = 20.0$, the range for which APE444 was designed with small fluctuations, the L-CNN model is superior in predicting the action and derivatives. Overall, the L-CNN performs well across the entire range from coarse to fine lattice spacing. Further details on the architecture search, on extended analysis of the results, on the generalization capabilities from restricted training ranges, on fine-tuning for different lattice sizes, on fine-tuning with instantons and on checking the lattice symmetries can be found in Appendix C.

## 4 CONCLUSIONS AND OUTLOOK

In this work we focused on describing in detail the challenging step to parametrize an FP action for the four-dimensional SU(3) gauge theory using L-CNNs and ML techniques. This allows us to compare with previous studies of the FP parametrization and also serves as a proof of concept that ML can be accurately used in this task. The end result is that the very expressive nature of L-CNNs enables us to find a much more accurate parametrization of the SU(3) FP action than previously possible. This success constitutes the first necessary step toward future Monte Carlo studies and ultimately toward the construction of a quantum perfect action without any lattice artifacts.

### ACKNOWLEDGMENTS

The authors wish to thank Anna Hasenfratz, Jim Hetrick, Gurtej Kanwar and Uwe-Jens Wiese for valuable discussions. This material is based upon work supported by the US National Science Foundation under Grant No. 2014150. KH and DM wish to thank the AEC and ITP at the University of Bern for their support. KH, AI and UW thank ECT* and the ExtreMe Matter Institute EMMI at GSI, Darmstadt, for support in the framework of the ECT*/EMMI Workshop "Machine learning in lattice field theory and beyond" during which this work has been developed. AI and DM acknowledge funding from the Austrian Science Fund (FWF) projects P 32446, P 34455 and P 34764. The computational results presented have been achieved in part using the Vienna Scientific Cluster (VSC) and computing resources at the University of Bern.

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

# A  FIXED POINT ACTION

Several different approaches are currently being followed to deal with the problems of critical slowing down and topological freezing. Simulations employing open boundary conditions in time (Lüscher & Schaefer, 2011) or huge master fields (Francis et al., 2020; Fritzsch et al., 2022) both circumvent topological freezing, but they do not address critical slowing down. Approaches using trivializing or normalizing flows (Lüscher, 2010) attempt to solve both problems by finding invertible maps from a simple probability distribution for the lattice configurations, which allows efficient sampling, to the target one. Recently, the use of machine-learning tools for parametrizing normalizing flows has roused anew attention in this approach (Albergo et al., 2019; Kanwar et al., 2020; Boyda et al., 2021b; Gerdes et al., 2023; Bacchio et al., 2023), however, these attempts are so far restricted to simple field theories, low dimensions or, in four-dimensional SU(3) gauge theories, to very small and coarse systems (Abbott et al., 2023). A complementary approach in order to solve both critical slowing down and topological freezing is by using a lattice action with no or highly suppressed lattice artifacts.

There is a long history of designing improved lattice actions to reduce discretization effects, bringing simulations at coarser lattice spacing into the scaling regime. One such program, Symanzik improvement (Symanzik, 1983a;b; Lüscher & Weisz, 1985a;b), removes lattice artifacts in some physical quantities order by order in the lattice spacing $a$. In a lattice gauge theory, this can be achieved, for example, by building a lattice action combining plaquettes and closed six-link loops. By construction, such an approach involves a perturbative expansion at weak coupling. A radically different approach makes use of renormalization group (RG) properties to design lattice actions where artifacts are removed completely to all orders. The construction of such *quantum perfect actions* is an extremely ambitious goal and is in general difficult to achieve. In asymptotically free theories, such as QCD, a constructive method can be designed based on the fixed point (FP) of RG transformations which yields lattice actions without lattice artifacts at the classical level, i.e., for on-shell quantities (Hasenfratz & Niedermayer, 1994). These so-called *classically perfect actions*, or *FP actions* in short, are in general expected to show suppressed lattice artifacts even at the quantum level. The FP action approach was used to study the O(3) nonlinear $\sigma$–model, SU(3) pure gauge theory, and full QCD, with promising indications of much-reduced cutoff dependence in Monte Carlo simulations (DeGrand et al., 1995b;a; 1996; Bietenholz & Wiese, 1996; Blatter et al., 1996; Blatter & Niedermayer, 1996; Niedermayer et al., 2001; Gattringer et al., 2004; Hasenfratz et al., 2004; 2005; 2009). However, the increased numerical cost of simulating FP gauge actions made it difficult at that time to draw firm conclusions on the level of improvement. Given the intervening dramatic increase in computing capability, this is no longer an obstacle and pushing the FP approach to higher accuracy has in principle become feasible.

The difficulty of implementing the FP program in practice stems from the fact that many of the FP properties are defined only implicitly without knowing the explicit form of the FP action. Moreover, the FP action in principle requires infinitely many loop operators in order to describe the infinitely many gauge link couplings generated through the renormalization group transformations (RGTs). This is not a problem *per se*, because reasonable choices of the RGT lead to FP actions which are local, i.e., for which the couplings decay exponentially with separation, and the RGT can in fact be designed to optimize this decay. For the SU(3) gauge theory this has been achieved by Blatter & Niedermayer (1996).

## A.1  THEORETICAL SETUP

The role of the Wilsonian renormalization group transformation (RGT) is to reduce the number of degrees of freedom of a particular physical system by integrating out fluctuations at high-energy scales while leaving the underlying physics at low-energy scales entirely intact (Wilson, 1971a;b). For a field theory regularized on a lattice, the lattice spacing is increased with each RGT step. Starting from a very fine lattice close to the continuum, for which any discretized action has negligible lattice artifacts, one can follow a chain of RGT steps leading to a very complicated lattice action on a coarse lattice describing the same low-energy physics. For SU($N_c$) lattice gauge theory where the underlying variables on the fine lattice $\Lambda = \{n \in \mathbb{N}^4\}$ are the gauge links $U_{n,\mu}$ with some lattice

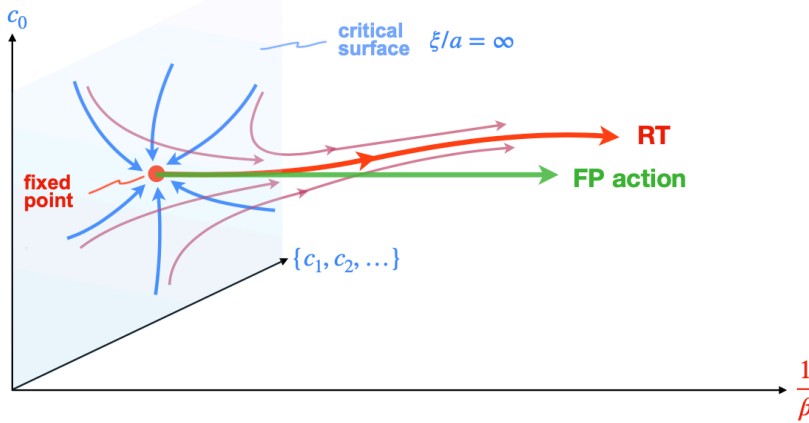

Figure 4: A sketch of the renormalization group flow and the renormalized trajectory (RT) in the infinite-dimensional coupling space, with the gauge coupling as the only relevant direction. The fixed point is on the critical surface $\beta \to \infty$ where $\xi/a = \infty$ for any physical scale $\xi$, with the values of the critical couplings $c_n^{\mathrm{FP}}$ determined by the specific form of the RG blocking. The FP action uses the same coupling values at finite $\beta$, tracking the RT very closely at weak coupling.

action $\mathcal{A}[U]$ and gauge coupling $\beta = 2N_c/g^2$, the RGT can be defined as

$$\exp(-\beta' \mathcal{A}'[V]) = \int \mathcal{D}U \exp(-\beta\{\mathcal{A}[U] + T[U,V]\}) \tag{2}$$

where the blocking kernel $T[U,V]$ is given by

$$T[U,V] = -\kappa \sum_{n_B,\mu} \left\{ \mathrm{ReTr}(V_{n_B,\mu} Q_{n_B,\mu}^\dagger) - \mathcal{N}_\mu^\beta \right\} \tag{3}$$

and defines the coupling between the fine links $U_{n,\mu}$ and the coarse links $V_{n_B,\mu}$ on the blocked coarse lattice $\Lambda_B = \{n_B \in \mathbb{N}^4\}$. The free parameter $\kappa$ can be optimized. The $Q_{n_B,\mu}$ variables are blocked links constructed from the underlying fine links $U_{n,\mu}$. The normalization term $\mathcal{N}_\mu^\beta$ guarantees that the partition function is invariant under the RGT, i.e., integrating Eq. (2) over the coarse gauge links with $\mathcal{D}V$ yields $Z(\beta') = Z(\beta)$. The form of the effective coarse action $\mathcal{A}'[V]$ and the couplings $\{g', c_0', c_1', \ldots\}$ are determined by the choice of the kernel $T[U,V]$. Under infinitesimal RGTs, the couplings map out a flow in the space of all possible gauge-invariant operators, as illustrated in Figure 4 by the light red trajectories.

For asymptotically free gauge theories, the only relevant coupling is the gauge coupling $g$ and the continuum is approached in the weak coupling limit $\beta \to \infty$. On the critical surface, where $\xi/a = \infty$, the irrelevant couplings $c_0, c_1, \ldots$ flow into a fixed point as shown in Figure 4. The FP couplings $\{c_0^{\mathrm{FP}}, c_1^{\mathrm{FP}}, \ldots\}$ are determined once the form of the RG blocking is prescribed. Slightly off the critical surface, the couplings first flow toward then away from the fixed point, approaching the renormalized trajectory (RT) which describes the flow starting from the FP in the relevant direction of the gauge coupling. Along the RT, the lattice theory is *quantum perfect*, with no lattice artifacts at all, because it is connected back to the continuum theory on the critical surface. The FP couplings define the so-called FP action $\mathcal{A}^{\mathrm{FP}}$. When it is used at finite values of $\beta$, it tracks the RT very closely at sufficiently weak coupling, cf. Figure 4. The FP action can be shown to be *classically perfect* (Hasenfratz & Niedermayer, 1994; DeGrand et al., 1995a), i.e., it has no lattice artifacts of $O(a^{2n})$ to all orders on field configurations fulfilling the equations of motions. Artifacts of $O(g^2 a^{2n})$ are, however, present but suppressed for small $g$.

As pointed out by Hasenfratz & Niedermayer (1994), the FP action $\mathcal{A}^{\mathrm{FP}}$ is implicitly given by the $\beta \to \infty$ limit of Eq. (2), namely by the saddle point equation

$$\mathcal{A}^{\mathrm{FP}}[V] = \min_{\{U\}} \left[ \mathcal{A}^{\mathrm{FP}}[U] + T[U,V] \right]. \tag{4}$$

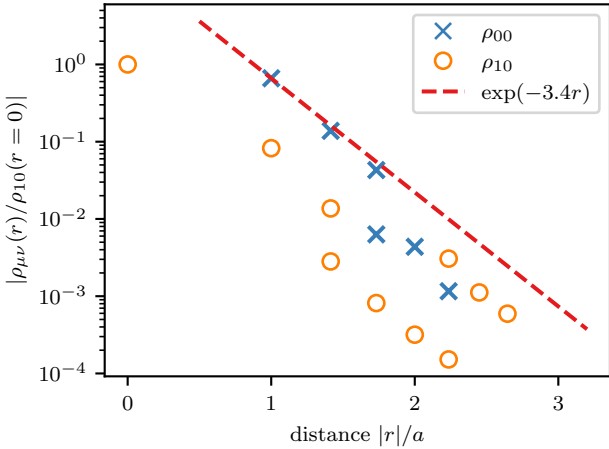

Figure 5: The leading couplings $\rho_{\mu\nu}(r)$ of the perturbative FP action, from Blatter & Niedermayer (1996). The blocking kernel $T[U, V]$ is designed to maximize the exponential decay of the couplings, with $\exp(-3.4r)$ shown as a visual guide.

For a fixed coarse configuration $V$, the minimization is over all possible fine configurations $U$, and the normalization term in the limit $\beta \to \infty$ becomes

$$\mathcal{N}_\mu^\infty = \max_{W \in \mathrm{SU}(N_c)} \left[ \mathrm{ReTr}(W Q_{n_B, \mu}^\dagger) \right] . \tag{5}$$

It is easy to see that the FP action has no lattice artifacts for field configurations fulfilling the equations of motion. It becomes apparent when considering the variation of the FP action using the chain rule,

$$\frac{\delta \mathcal{A}^{\mathrm{FP}}[V]}{\delta V} = \left[ \frac{\delta}{\delta U}(\mathcal{A}^{\mathrm{FP}}[U] + T[U, V])\frac{\delta U}{\delta V} + \frac{\delta T[U, V]}{\delta V} \right]_{U_{\min}} \tag{6}$$

where $U_{\min}$ is the configuration minimizing the right-hand side of Eq. (4). For a classical coarse configuration $V$ one has

$$\frac{\delta \mathcal{A}^{\mathrm{FP}}[V]}{\delta V} = 0 \quad \Rightarrow \quad \left. \frac{\delta T[U, V]}{\delta V} \right|_{U_{\min}} = 0 \tag{7}$$

since $U_{\min}$ minimizes the sum $\mathcal{A}^{\mathrm{FP}}[U] + T[U, V]$. Hence $T[U, V]$ takes its minimum value, namely zero. This in turn forces

$$\mathcal{A}^{\mathrm{FP}}[V] = \mathcal{A}^{\mathrm{FP}}[U_{\min}], \quad \left. \frac{\delta \mathcal{A}^{\mathrm{FP}}[U]}{\delta U} \right|_{U_{\min}} = 0, \tag{8}$$

meaning the minimizing configuration $U_{\min}$ is also classical and the FP action value is unchanged in the minimizing step. This can be iterated until one reaches an arbitrarily fine classical solution with the correct continuum action value. In particular, the FP action allows for exact instanton solutions at finite lattice spacing (Blatter et al., 1996) and the exact FP equation therefore preserves topology on the lattice. Note, however, that this is not necessarily true for the RGT step. Starting from a fine configuration $U$, which is a classical solution, the resulting blocked configuration $V$ might not automatically be one as well. In fact, this can directly be seen by blocking analytical instanton solutions with a small radius in lattice units such that the instanton properties are lost on the coarse configuration. This process of instantons falling through the lattice is discussed further in Sec. A.2.

A crucial question concerns the locality of the FP action or, more generally, the action $\mathcal{A}'$ in Eq. (2). In order to guarantee universality, the couplings must fall off exponentially in the separation between fields. One can design RGTs which force an exponential decay and find the one maximizing it, so that beyond some separation the couplings are small enough to be negligible and in practical applications can be omitted. Some guidance for a good choice of blocking kernel can be provided

perturbatively (DeGrand et al., 1995a;b). At weak coupling (and with some gauge fixing), the FP action can be expanded in terms of the gauge potential $A_\mu^a(x)$, only keeping terms up to quadratic order in the potential. The resulting action can be expressed in terms of couplings $\rho_{\mu\nu}(r)$ for fields $A_\mu(x)$ and $A_\nu(y)$ at separation $r = |x - y|$. An optimal choice of the blocking and the RGT w.r.t. locality was found in Blatter & Niedermayer (1996). Figure 5 shows the corresponding largest perturbative couplings which fall off exponentially in magnitude with $\sim \exp(-3.4r)$. It is this RGT which we employ in our work.

The FP action $\mathcal{A}^{\text{FP}}$ and its properties are defined only implicitly through the FP Eq. (4), where $\mathcal{A}^{\text{FP}}$ appears on both sides. The FP equation is therefore iterative: On the right-hand side of Eq. (4), the value of $\mathcal{A}^{\text{FP}}[U]$ can be determined through a second minimization over even finer gauge configurations $U'$, and so on, until we reach a configuration so smooth that any (reasonable) lattice discretization of the continuum Yang-Mills gauge action can be used to calculate the inception value of the action. In practice, instead of iterating the FP equation, one can shortcut the procedure and, for sufficiently smooth fine configurations, make use of existing approximate parametrizations of $\mathcal{A}^{\text{FP}}[U]$. Previous parametrizations include linear combinations of plaquette, rectangular, and parallelogram loops with various powers of their traces (Blatter & Niedermayer, 1996), or combinations of thin-link and smeared-link plaquette traces $u_{x,\mu\nu}$ and $w_{x,\mu\nu}$ with various powers of the form

$$\mathcal{A}^{\text{FP}}[V] = \frac{1}{N_c} \sum_{x,\mu<\nu} \sum_{k,l} p_{kl} u_{x,\mu\nu}^k w_{x,\mu\nu}^l \tag{9}$$

with optimized coefficients $p_{kl}$ (Niedermayer et al., 2001). While this parametrization ansatz is already very general and flexible, in practice one is restricted to a rather small set of $O(20 - 30)$ parameters. In this paper, we take a different approach using L-CNNs and ML in connection with the FP data from Eq. (4) in order to explore a much larger space of possible actions, with the goal of finding a more accurate approximation than previously feasible — that is the parametrization challenge which we address in this paper.

## A.2 FIXED POINT DATA

To parametrize the FP action accurately requires a large set of data. In this section we describe how this data is obtained on the basis of Eq. (4). In this work, most of the FP data stems from Monte Carlo ensembles generated using the Wilson gauge action at various couplings $\beta_{\text{wil}}$. As such, $\beta_{\text{wil}}$ simply serves as a proxy for the size and characteristics of the gauge field fluctuations. For each coarse configuration $V$ one needs to find the minimizing fine configuration $U_{\text{min}}$ on the right-hand side of Eq. (4) which then yields the value $\mathcal{A}^{\text{FP}}[V]$.

As described in the previous section, for practical reasons one employs a parametrization of $\mathcal{A}^{\text{FP}}[U]$ for the minimization procedure and the question arises how this approximation affects the true value $\mathcal{A}^{\text{FP}}[V]$. Since the RG blocking increases the lattice spacing by a factor of two in each RGT step, the action density on the fine configuration $U$ is at least a factor of 16 smaller than on the coarse configuration $V$, and in practice is even smaller, because of the sizable positive contribution from the blocking kernel $T[U, V]$. In Figure 6 we show the two contributions $T[U, V]$ and $\mathcal{A}^{\text{FP}}[U]$ to $\mathcal{A}^{\text{FP}}[V]$ averaged over $4^4$ lattice ensembles generated at the indicated coupling $\beta_{\text{wil}}$ and we find that for $\beta_{\text{wil}} \gtrsim 5.5$ the action density for $\mathcal{A}^{\text{FP}}[U]$ is about a factor $\gtrsim 30$ smaller than the one for $\mathcal{A}^{\text{FP}}[V]$. (Note that in the figure the action density for $\mathcal{A}^{\text{FP}}[U]$ is normalized to the coarse lattice volume.) Hence, for the very smooth fine configurations, any reasonably good approximation to $\mathcal{A}^{\text{FP}}[U]$ can be used on the right-hand side and in practice we employ the existing APE444 parametrization. This action is constructed in such a way that the couplings of the FP action in the quadratic approximation are reproduced (Blatter & Niedermayer, 1996) while explicitly maintaining the Symanzik "on-shell" conditions to $O(a^2)$ (Niedermayer et al., 2001), and it therefore is a very good approximation on sufficiently smooth configurations. From Figure 6 we can estimate the error on $\mathcal{A}^{\text{FP}}[V]$ induced by using the APE444 parametrization on the right-hand side. Considering the worst case $\beta_{\text{wil}} = 5.0$, for the minimizing configurations we find action densities $\lesssim 0.5$ corresponding to $\beta_{\text{wil}} \gg 20.0$. From the top plot in Figure 3 we find that for the APE444 parametrization the relative action error is $\lesssim 0.3\%$ inducing an error of $\lesssim 0.17\%$ on $\mathcal{A}^{\text{FP}}[V]$ for configurations at $\beta_{\text{wil}} = 5.0$ and far less than $0.1\%$ already at $\beta_{\text{wil}} = 6.0$. The accuracy of $\mathcal{A}^{\text{APE444}}[U]$ can of course also be checked by further minimization over $U'$.

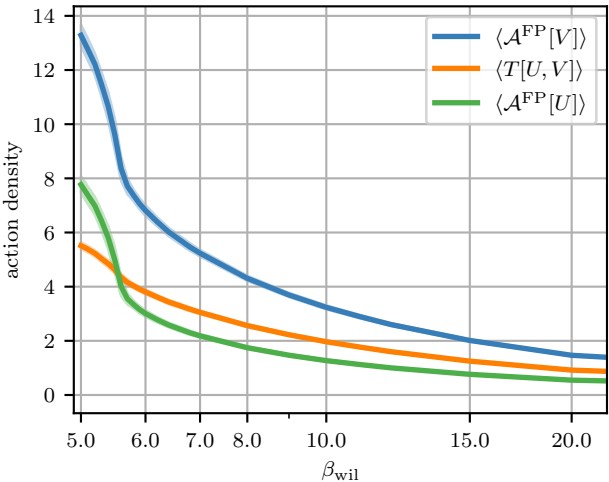

Figure 6: Fixed point action density as a function of $\beta_{\mathrm{wil}}$ on a $4^4$ lattice. We show the ensemble-averaged FP action $\mathcal{A}^{\mathrm{FP}}[V]$, the blocking kernel $T[U,V]$, and the parametrized FP action $\mathcal{A}^{\mathrm{FP}}[U]$ used on the right-hand side of Eq. (4), normalized to the coarse lattice volume. The mean values are obtained by averaging over the ensemble at a given $\beta_{\mathrm{wil}}$. The shaded regions indicate the standard deviation.

Another potential error on the FP data $\mathcal{A}^{\mathrm{FP}}[V]$ may originate from inaccurate minimization of the right-hand side of Eq. (4). The minimization on each configuration starts from an initial random fine configuration $U$ and then sequentially updates each link $U_{n,\mu}$ with an adaptive rotation in color space. Each iteration corresponds to a pass through the entire volume. We show two typical examples of this minimizing procedure in Fig. 7 for two coarse configurations $V$ on $8^4$ volume generated with $\beta_{\mathrm{wil}} = 6.0$ (top plot) and $\beta_{\mathrm{wil}} = 5.4$ (bottom plot). As shown in the figure, on smoother configurations at $\beta_{\mathrm{wil}} = 6.0$ the minimization converges quickly, while on rougher configurations at $\beta_{\mathrm{wil}} = 5.4$ as expected it takes somewhat longer to reach a similar level of convergence. In any case, we see from the illustrations that even in those cases, the error on the value of the FP action $\mathcal{A}^{\mathrm{FP}}[V]$ is negligible. Note that the minimization procedure is the most expensive step in generating the FP data. This is because the update of a single link $U_{n,\mu}$ contributes both to $\mathcal{A}^{\mathrm{APE444}}[U]$ and several blocked links $Q_{n_B,\nu}[U]$ in a complicated way which requires the expensive recalculation of many intermediate quantities and the resulting contributions.

The action value $\mathcal{A}^{\mathrm{FP}}[V]$ is only one datum of information for each coarse configuration $V$. However, the FP Eq. (4) contains much more information which can be extracted from the derivatives w.r.t. the gauge links (Niedermayer et al., 2001). Since the first term on the right-hand side of Eq. (6) vanishes for the minimizing configuration $U_{\mathrm{min}}$, the derivative can be determined solely from the blocking kernel evaluated on the minimizing configuration. To be explicit, one has

$$\frac{\delta \mathcal{A}^{\mathrm{FP}}[V]}{\delta V_{x,\mu}^a} = \left.\frac{\delta T[U,V]}{\delta V_{x,\mu}^a}\right|_{U_{\mathrm{min}}} = -\kappa \mathrm{ReTr}(it^a V_{x,\mu} Q_{x,\mu}^\dagger), \tag{10}$$

with $t^a$ the generators of SU(3) and the blocked links $Q_{x,\mu}$ built from the minimizing configuration $U_{\mathrm{min}}$. The derivative notation concretely means

$$\frac{\delta f(V)}{\delta V_{x,\mu}^a} = \lim_{\epsilon \to 0} \frac{1}{\epsilon}\left(f(e^{i\epsilon X}V) - f(V)\right), \ \ X(y,\nu) = t^a \delta_{xy}\delta_{\mu\nu} \tag{11}$$

for any scalar function $f(V)$. Each coarse gauge configuration $V$ on a $L^4$ lattice therefore generates $4 \times L^4 \times (N_c^2 - 1)$ data of derivatives, one for each link and color index. This is a large amount of information which is very valuable in the parametrization process as it tightly constrains the form of the FP action. For later convenience, we combine the derivatives in the form

$$D_{x,\mu}^{\mathrm{FP}} = \sum_a t^a \frac{\delta \mathcal{A}^{\mathrm{FP}}[V]}{\delta V_{x,\mu}^a} \tag{12}$$

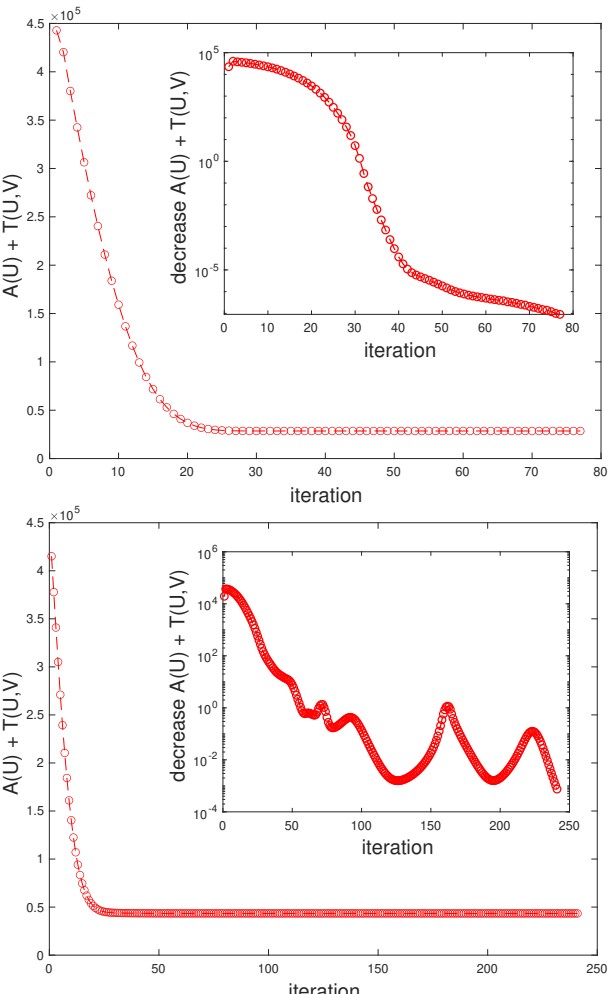

Figure 7: Examples of minimization on $8^4$ lattice configurations with $\beta_{\mathrm{wil}} = 6.0$ (top) and 5.4 (bottom). The insets show the decrease of $\mathcal{A}[U] + T[U, V]$ in each iteration.

which makes them independent of the choice of basis for the generators.

Gauge invariance of the FP action means the derivatives $D_{x,\mu}^{\mathrm{FP}}$ are not independent, which yields a very useful consistency check. Under an infinitesimal transformation of the links $V'_{x,\mu} = R_x V_{x,\mu} R^{\dagger}_{x+\hat{\mu}}$ with $R_x = \exp(i\alpha_x^a t^a)$, the action being unchanged forces

$$\sum_{x,\mu} \mathrm{Tr}\left[(\mathcal{D}_{\mu}^{F}\alpha_x)D_{x,\mu}^{\mathrm{FP}}[V]\right] = 0, \tag{13}$$

with $\alpha_x = \alpha_x^a t^a$ and the gauge covariant forward finite difference $\mathcal{D}_{\mu}^{F}\alpha_x = V_{x,\mu}\alpha_{x+\hat{\mu}}V^{\dagger}_{x,\mu} - \alpha_x$. After summation by parts, this is equivalent to the condition

$$\sum_{x,\mu} \mathrm{Tr}\left[\alpha_x \mathcal{D}_{\mu}^{B} D_{x,\mu}^{\mathrm{FP}}[V]\right] = 0, \tag{14}$$

with the gauge covariant backward finite difference defined as $\mathcal{D}_{\mu}^{B}G_x = G_x - V^{\dagger}_{x-\hat{\mu},\mu}G_{x-\hat{\mu}}V_{x-\hat{\mu},\mu}$ for a matrix-valued field $G_x$. Since Eq. (14) has to be satisfied for all possible $\alpha_x$, this becomes a local condition

$$\sum_{\mu} \mathcal{D}_{\mu}^{B} D_{x,\mu}^{\mathrm{FP}}[V] = 0 \tag{15}$$

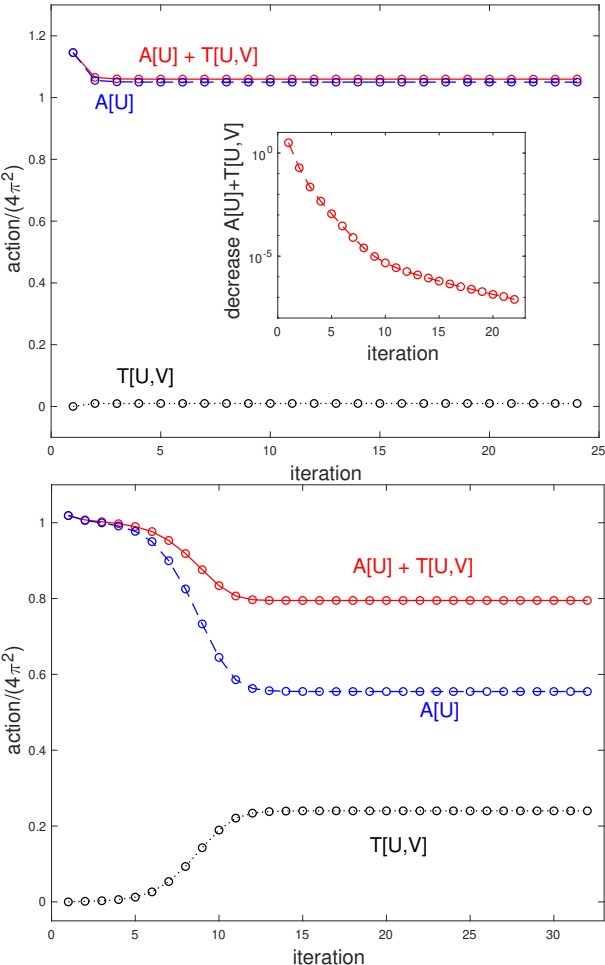

Figure 8: Minimization of instanton configurations. In the upper panel, an instanton of size $\rho/a = 3.0$ on a $16^4$ lattice persists after being blocked to the $8^4$ lattice. The lower panel shows an instanton with $\rho/a = 1.5$ which is too small to survive the RG blocking, i.e., the instanton falls through the lattice.

at each $x$ to be true for exactly gauge invariant actions. We note that Eq. (15) is a consequence of Noether's second theorem applied to the FP action.

In our approach, we compute the FP derivatives using Eq. (10), relying on the fact that $U$ is a (local) minimum of the right-hand side of the FP equation. Since the numerical minimization procedure to determine the fine configuration $U$ can only yield approximate minima, we may check, for each coarse configuration $V$, how closely the numerically obtained FP derivatives satisfy this requirement. This allows us to directly assess the quality of the minimizing configuration and the FP action data. In practice, we find that the consistency check is satisfied up to the accuracy achieved in the minimization.

In addition to Monte Carlo ensembles generated with the Wilson gauge action, we can also examine the FP action for instanton lattice configurations. Taking as input a fine instanton configuration with some chosen value of instanton radius $\rho$, we produce a coarse configuration $V$ using the RG blocking. If the topological properties are intact on the coarse side, the FP action should be unchanged by minimization, reproducing an instanton solution on the fine side. We see tests of this in Fig. 8. Starting from a fine instanton solution on a $16^4$ volume, the coarse $8^4$ configuration $V$ is produced via RG blocking and then fed into the minimization procedure. The upper plot is for an instanton originally of radius $\rho/a = 3.0$; under minimization, the action is essentially unchanged, with a very small contribution from the blocking kernel $T[U, V]$, meaning the blocked configuration is also an

instanton solution. The inset shows the rapid convergence of $\mathcal{A}[U] + T[U, V]$ in the minimization. The lower plot is for an instanton originally of radius $\rho/a = 1.5$; once RG-blocked, the instanton is lost, as $T[U, V]$ becomes much larger during minimization and the minimized total action $\mathcal{A}[U] + T[U, V]$ is below the continuum value $4\pi^2$, i.e., the topological features are lost because the instanton can non longer be resolved at the level of the coarse lattice spacing $a' = 2a$. Note that with the RGT-III blocking employed in this work, instantons fall through the lattice for radii $\rho/a' \lesssim 0.85$. In order to embrace this specific classical property of the FP action, we generate a set of coarse configurations through blocking fine instanton configurations with $\rho/a$ ranging from 1.1 to 3.0. The corresponding FP action values and derivatives provided by the minimization form part of the FP training data set.

## B  MACHINE LEARNING MODEL

Machine learning is being applied across a vast array of fields (LeCun et al., 1998; 2015; Mehta et al., 2019; Feickert & Nachman, 2021; Boyda et al., 2022; Shanahan et al., 2022). Focused more specifically on lattice field theory, it has been used to in a range of topics, including the identification of phase transitions and their underlying critical exponents (Bachtis et al., 2020), the generation of decorrelated Markov chains through normalizing (Kanwar et al., 2020) or trivializing (Bacchio et al., 2023) flow transformations, inverting renormalization group transformations in scalar field theory (Bachtis et al., 2022), the finite-temperature deconfinement phase transition in SU(2) and SU(3) pure gauge theory (Boyda et al., 2021a; Gerasimeniuk et al., 2022), preconditioning of lattice Dirac operators (Lehner & Wettig, 2023a;b), and the connection between machine learning diffusion models and stochastic quantization of field theories through Langevin dynamics (Wang et al., 2023). A recent review of some of this work can be found in Kanwar (2024). In our context, we need a tool to parametrize a lattice action in a highly general form, maintaining exact gauge invariance. The necessary architecture has already been developed by Favoni et al. (2022) with the lattice gauge equivariant convolutional neural network (L-CNN).

### B.1  GAUGE EQUIVARIANT NETWORK LAYERS

The input to the L-CNN network (Favoni et al., 2022) is a set of gauge configurations $U_{x,\mu}$, which under a gauge transformation change as $U'_{x,\mu} = \Omega_x U_{x,\mu} \Omega^\dagger_{x+\hat{\mu}}$, with $\Omega_x \in$ SU(3). From the gauge links, we build untraced plaquette variables

$$U_{x,\mu\nu} = U_{x,\mu} U_{x+\hat{\mu},\nu} U^\dagger_{x+\hat{\nu},\mu} U^\dagger_{x,\nu} = \boxed{\phantom{x}}, \tag{16}$$

which gauge transform locally as $U'_{x,\mu\nu} = \Omega_x U_{x,\mu\nu} \Omega^\dagger_x$. We refer to generic variables with local gauge transformations as $W_{x,a}$ with channel index $1 \leq a \leq N_{\text{ch}}$. Gauge equivariant convolutions of these variables (the "channels") are built through parallel transport via gauge links:

$$W_{x,a} \rightarrow \sum_{b,\mu,k} \omega_{a,b,\mu,k} U_{x,k\cdot\hat{\mu}} W_{x+k\cdot\hat{\mu},b} U^\dagger_{x,k\cdot\hat{\mu}}, \tag{17}$$

with $\omega_{a,b,\mu,k}$ the convolution weights, channel indices $1 \leq a \leq N_{\text{ch,out}}$ and $1 \leq b \leq N_{\text{ch,in}}$, and $-(K-1) \leq k \leq (K-1)$ with $K$ the kernel size. The parallel transporters $U_{x,k\cdot\hat{\mu}}$, which start at $x$ and end at $x + k \cdot \hat{\mu}$, are the products of consecutive gauge links along the path. Products of locally transforming variables are constructed in a bilinear layer

$$W_{x,a} \rightarrow \sum_{b,c} \alpha_{a,b,c} W_{x,b} W'_{x,c} \tag{18}$$

with parameters $\alpha_{a,b,c}$ and channel indices in the ranges $1 \leq b \leq N_{\text{in},1}, 1 \leq c \leq N_{\text{in},2}$ and $1 \leq a \leq N_{\text{out}}$, a crucial point being that gauge covariance is maintained exactly as the product is of variables at the same lattice site. For the L-CNN models used in this work, we use a combination of the convolutional and bilinear layer (a *bilinear convolution*), which can be expressed as

$$W_{x,a} \rightarrow \sum_{b,c,k,\mu} \omega_{a,b,c,k,\mu} W_{x,b} U_{x,k\cdot\hat{\mu}} W_{x+k\cdot\hat{\mu},c} U^\dagger_{x,k\cdot\hat{\mu}}, \tag{19}$$

where $\omega_{a,b,c,k,\mu}$ are real-valued weights and $1 \leq a \leq N_{\text{out}}$ (output channels), $1 \leq c, b \leq N_{\text{in}}$ (input channels) and $-(K-1) \leq k \leq (K-1)$. We also note that each bilinear convolutional layer considers both orientations of a particular input variable (e.g. both $W_{x,i}$ and $W_{x,i}^{\dagger}$), which effectively doubles the number of input channels, and a residual term. The number of trainable parameters associated with Eq. (19) is given by $(2D(K-1)N_{\text{in}}+1) \cdot (2N_{\text{in}}+1) \cdot N_{\text{out}}$ with $D$ the dimension of the lattice.

### B.2 PARAMETRIZING ACTIONS USING L-CNNs

The parametrization of the action uses the ansatz presented in Eq. (1). We consider prefactor actions of the form

$$\mathcal{A}_x^{\text{pre}}[V] = \frac{1}{N_c} \sum_{\mathcal{C}} \sum_{m=1}^{M} c_{\mathcal{C}}^{(m)} \left[ \text{ReTr}(\mathbb{1} - U_{x,\mathcal{C}}) \right]^m , \tag{20}$$

where we sum over a set of Wilson loops $U_{x,\mathcal{C}}$ (specifically plaquettes, rectangles, chairs, and parallelograms) starting at the lattice site $x$ and $c_{\mathcal{C}}^{(m)}$ are real-valued coefficients. By construction, the prefactor action in Eq. (20) is ultralocal, with zero coupling beyond some separation. Additionally, there are constraints on the coefficients $c_{\mathcal{C}}^{(m)}$ which ensure positivity. Particular choices for the coefficients guarantee that the prefactor action approaches the Yang-Mills action smoothly (normalization) and is optionally improved to some particular order (Symanzik improvement). Suitable choices for the prefactor action are the Wilson action (which only consists of the linear plaquette term) or the Symanzik improved action (linear plaquette and rectangle contributions). The specific form of Eq. (20) also allows for the fixed point action parametrizations considered in Blatter & Niedermayer (1996), specifically the type IIIa, IIIb and IIIc parametrizations, which include all terms except chairs up to order $M = 4$. If the set of Wilson loops includes plaquettes, rectangles, parallelograms, and chairs, the normalization condition is

$$c_{\text{pl}}^{(1)} + 8c_{\text{rt}}^{(1)} + 8c_{\text{pg}}^{(1)} + 16c_{\text{ch}}^{(1)} = 1, \tag{21}$$

while the Symanzik conditions that can be imposed are Lüscher & Weisz (1985a)

$$c_{\text{rt}}^{(1)} - c_{\text{pg}}^{(1)} - c_{\text{ch}}^{(1)} = -\frac{1}{12}, \qquad c_{\text{pg}}^{(1)} = 0. \tag{22}$$

The most frequently used Symanzik improved gauge action sets $c_{\text{ch}}^{(1)} = 0$, combining only plaquettes and rectangles with $c_{\text{rt}}^{(1)} = -1/12$ and $c_{\text{pl}}^{(1)} = 5/3$. Note that the parametrized FP action of Blatter & Niedermayer (1996) set $c_{\text{ch}}^{(1)} = 0$, but included parallelogram loops as well. While the prefactor part of $\mathcal{A}^{\text{L-CNN}}[V]$ is designed to provide a good approximation to $\mathcal{A}^{\text{FP}}[V]$ for smooth gauge fields, we use the term $N_x[V]$ to deal with coarse configurations. We represent $N_x[V]$ as the real trace of a stack of $N_{\text{layer}} \geq 1$ bilinear convolutional layers. The output of the L-CNN is thus a linear combination of Wilson loops of various sizes and therefore local. We regularize the output of the model such that the difference $N_x[V] - N_x[\mathbb{1}]$ vanishes in the vacuum for $V_{x,\mu} = \mathbb{1}$.[1] Furthermore, since the L-CNN can be written as a linear combination of Wilson loops, a naive continuum expansion using $V_{x,\mu} = \exp(iaA_\mu(x))$ yields

$$N_x[V \to \mathbb{1}] - N_x[\mathbb{1}] \simeq O(a^2). \tag{23}$$

The leading order term of the parametrized action is thus

$$\mathcal{A}^{\text{L-CNN}}[V \to \mathbb{1}] \simeq \mathcal{A}^{\text{pre}}[V](1 + b^{(1)}O(a^2)). \tag{24}$$

Our chosen ansatz therefore guarantees the correct continuum behavior of the action.

The positivity requirement is realized if the prefactor action is positive everywhere and if the coefficients $b^{(n)}$ are chosen appropriately. For example, we may use $b^{(n)} = 1/n!$ which allows us to write the parametrized action as

$$\mathcal{A}_{(\exp)}^{\text{L-CNN}}[V] = \sum_x \mathcal{A}_x^{\text{pre}}[V] \exp(N_x[V] - N_x[\mathbb{1}]), \tag{25}$$

---

[1] This also holds for gauge equivalent vacuum configurations $V_{x,\mu} = \Omega_x \Omega_{x+\hat{\mu}}^{\dagger}$.

which is positive for all gauge configurations. Another simple choice is to truncate at order $n = 1$:

$$\mathcal{A}^{\text{L-CNN}}_{(\text{lin})}[V] = \sum_x \mathcal{A}^{\text{pre}}_x[V](1 + N_x[V] - N_x[\mathbb{1}]). \tag{26}$$

We note however that this ansatz is not manifestly positive.

### B.3 Training

In the present context, we train the L-CNN using ensembles of gauge configurations $\{V_i\}$, for which the values of the fixed point action and associated derivatives have been obtained through minimization as in Eqs. (4) and (10). The output of the L-CNN is $\mathcal{A}^{\text{L-CNN}}[V_i]$. The predicted derivatives $D^{\text{L-CNN}}_{x,\mu}[V] = \sum_a t^a \delta \mathcal{A}^{\text{L-CNN}}[V]/\delta V^a_{x,\mu}$ (analogous to Eq. (12)) are calculated exactly through backpropagation: instead of varying the output of the neural network with respect to the parameters of the model to minimize a loss function, we calculate the derivative of the network output with respect to the input variables, the gauge links. With this information, the loss function $\mathcal{L}$ for the L-CNN is a combination of

$$\mathcal{L}_1 = \frac{1}{L^4 N_{\text{cfg}}} \sum_{i=1}^{N_{\text{cfg}}} |\mathcal{A}^{\text{FP}}[V_i] - \mathcal{A}^{\text{L-CNN}}[V_i]|,$$

$$\mathcal{L}_2 = \frac{1}{32 L^4 N_{\text{cfg}}} \sum_{i=1}^{N_{\text{cfg}}} \sum_{x,\mu} \text{Tr}\left[(D^{\text{FP}}_{x,\mu}[V_i] - D^{\text{L-CNN}}_{x,\mu}[V_i])^2\right],$$

$$\mathcal{L} = w_1 \mathcal{L}_1 + w_2 \mathcal{L}_2, \tag{27}$$

where $N_{\text{cfg}}$ is the number of configurations in the data set. The weights $w_{1,2}$ for the loss function are hyperparameters of the model. Typically, we use $w_1 = 0.1$ and $w_2 = 1$. The model is trained by minimizing $\mathcal{L}$ using the *AdamW* optimizer. Note that $\mathcal{L}_2$ contains the group derivatives $D^{\text{L-CNN}}_{x,\mu}$ of the model which we compute by relating them to matrix-valued Wirtinger derivatives (see Appendix D for details). Unless stated otherwise, we use single precision for floating-point arithmetic during training and testing.

The data used to train and evaluate the network are SU(3) gauge ensembles on volumes $4^4$, $6^4$, and $8^4$ with the Wilson gauge action and bare gauge couplings $\beta_{\text{wil}}$ ranging from 5.0 to 100.0, with more dense spacing in $\beta_{\text{wil}}$ at the stronger coupling end. Each member of these ensembles represents a possible coarse configuration $V$ in Eq. (4), the minimization procedure begins with a random starting fine configuration $U$ and a parametrization of $\mathcal{A}^{\text{FP}}[U]$ appropriate for smooth gauge links. Here, we use the APE444 parametrization (Blatter & Niedermayer, 1996). Minimizing $\mathcal{A}^{\text{FP}}[U] + T[U,V]$ by adaptively updating of the links $U$ produces sets of fine configurations with matching volumes $8^4$, $12^4$ and $16^4$. Each ensemble consists of 200 saved configurations equally spaced from Markov chains of length $10^6$, the ensembles are split into 80% training, 10% validation, and 10% test data.

## C Further details on results

### C.1 Architecture search

The flexibility of the L-CNN architecture allows for a large variation of the network hyperparameters, namely the number of bilinear convolution layers, the number of channels, and the kernel size for convolutions. To gain some insight as to the optimal choices for these hyperparameters, we train a large set of models on the same data set, gauge ensembles with lattice volume $4^4$ generated with the Wilson gauge action and bare coupling $\beta_{\text{wil}}$ from 5.0 to 10.0, for which minimization was first done to find the corresponding values of the FP action and derivatives. In the L-CNN models, we use the local Wilson action density as the prefactor $\mathcal{A}^{\text{pre}}_x[V]$. Details about the various architectures are shown in Table 1, where we list the number of bilinear convolutional layers, and their associated kernel sizes and output channels. We also provide the number of trainable parameters. As detailed at the end of Section B.1, the number of parameters for each bilinear convolution grows quickly with the number of channels, the kernel size, and the number of dimensions. For each unique architecture of the thirteen listed in the table, we use both Eqs. (25) and (26) and train each architecture five times using random initial weights. This amounts to a total of 130 unique models.

Table 1: Architecture details of the hyperparameter scan. All architectures use the Wilson action density as a prefactor action and use clover leaf plaquettes (24 input channels). After the last convolution, we take the real part of the trace and use a final linear layer to map the remaining channels to a single real number.

| Layers | Kernel Sizes | Channels | Parameters |
|---|---|---|---|
| 1 | 1 | 4 | 9.61K |
| | 2 | 8 | 170K |
| | 2 | 16 | 340K |
| 2 | 1, 1 | 4, 8 | 10.3K |
| | 2, 1 | 8, 16 | 174K |
| | 2, 2 | 16, 12 | 454K |
| 3 | 2, 1, 1 | 4, 4, 8 | 85.8K |
| | 2, 2, 1 | 8, 8, 16 | 194K |
| | 2, 2, 1 | 12, 24, 24 | 443K |
| | 2, 2, 1 | 16, 16, 32 | 527K |
| 4 | 2, 1, 1, 1 | 8, 8, 16, 32 | 212K |
| | 2, 2, 1, 1 | 16, 16, 16, 32 | 544K |
| | 2, 2, 2, 1 | 16, 24, 24, 32 | 1.15M |

The firm indication is that L-CNN models with three layers, cumulative kernel sizes of five, and cumulative number of channels approximately 60, are highly accurate, predicting the FP action with an error well below 1%. Although not explicitly shown in Fig. 2, we remark that we find little difference in the choice of function that is used to combine the prefactor action with the regularized L-CNN: Both the exponential and linear functions in Eqs. (25) and (26) show virtually the same performance across all tested architectures. Since the exponential function is manifestly positive and thus more likely to produce strictly positive parametrizations, we deem it the more suitable choice for further studies. We note that results for models with up to three layers were obtained using 400 training epochs, whereas models with four layers required 1000 epochs for convergence. During the training phase of models with four layers, we encountered a single outlier, which did not converge.

The broad scan allows us to narrow the search for the optimal L-CNN, for which training can be extended to a larger number of epochs to ensure convergence. We can also avail of previous studies of the FP action for SU(3) gauge theory, where the accuracy of those parametrizations provides a baseline. The older study (Blatter & Niedermayer, 1996) used the ansatz as in Eq. (20), with plaquette, rectangle and parallelogram loops, and powers up to $M = 4$, with the coefficients $c_C^{(m)}$ determined through $\chi^2$ minimization. Borrowing their nomenclature we refer to this parametrization as IIIc-4 in the figures. The later study (Niedermayer et al., 2001) used a very different ansatz as in Eq. (9), with powers of plaquettes of original and smeared gauge links, with the smearing sensitive to the local fluctuations of the gauge links. This yielded two parametrizations, one designed to be accurate on smooth gauge configurations close to the continuum (denoted APE444 in our figures) and a second to be used on rough configurations with a lattice spacing as large as 0.35 fm (referred to as APE431).

Motivated by the hyperparameter scan, we decide on training architectures with three bilinear convolutional layers, using kernel sizes $\{2, 2, 1\}$ and output channels $\{12, 24, 24\}$ respectively. To improve the behavior in the continuum $\beta_{\text{wil}} \to \infty$ we opt for a prefactor action of type IIIc-4 and extend the range of training data to $5.0 \leq \beta_{\text{wil}} \leq 20.0$ on $4^4$ lattices. Furthermore, we may consider the parameters of the prefactor in Eq. (20) to be adjusted during training while ensuring that the normalization and Symanzik conditions remain satisfied. Instead of using random initialization, we set the coefficients $c_C^{(m)}$ to the values originally found in Blatter & Niedermayer (1996). Thus, both the prefactor coefficients and the weights of the L-CNN are optimized during training. We train these models using multiple random initializations for 800 epochs. Results of employing finetuning to further improve our models are discussed in Appendix C.5. Figures 3 to 12, which we discuss in detail in the following section, are produced with our best model found through this training procedure including finetuning on instantons.

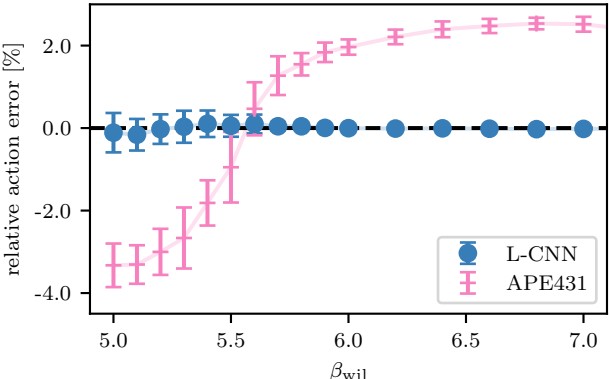

Figure 9: Comparison of different parametrizations of the FP action evaluated on MC ensembles from $\beta_{\text{wil}} = 5.0$ to $\beta_{\text{wil}} = 7.0$ on a $4^4$ lattice. We plot the relative linear deviations from the numerical fixed point action data for our best L-CNN model and the APE431 parametrization.

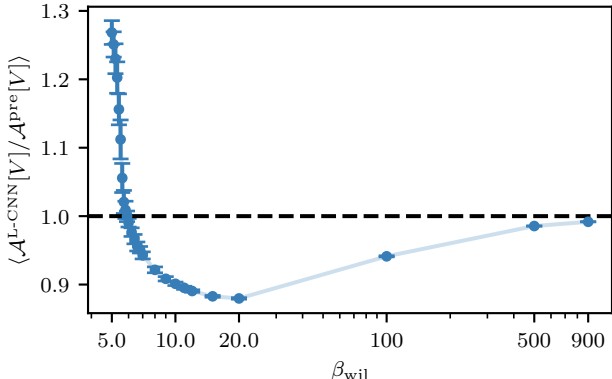

Figure 10: Ratio of our best L-CNN model $\mathcal{A}^{\text{L-CNN}}[U]$ and its associated prefactor action $\mathcal{A}^{\text{pre}}[U]$ (in this case, a learned IIIc action) as a function of $\beta_{\text{wil}}$ on a $4^4$ lattice. By construction, the L-CNN model approaches the prefactor in the limit of smooth configurations $\beta_{\text{wil}} \to \infty$.

## C.2 DETAILED RESULTS

To amplify the superiority of the trained network, we show in Fig. 9 the difference between predicted and actual FP action values for APE431 (designed for coarse lattices) and L-CNN in the range $5.0 \leq \beta_{\text{wil}} \leq 7.0$. The difference changes sign for APE431 as we scan across bare coupling, the L-CNN model gives a visibly much more accurate prediction. The effect of the model can be drawn out as shown in Fig. 10 through the ratio of the L-CNN output $\mathcal{A}^{\text{L-CNN}}[V]$ to the prefactor $\mathcal{A}^{\text{pre}}[V]$, which varies up to $\sim 30\%$ on the coarsest gauge ensembles, approaching 1 in the continuum limit $\beta_{\text{wil}} \to \infty$.

Because the FP derivatives represent a volume-sized amount of information for each gauge configuration, the distributions of the error $D_{x,\mu,a}^{\text{FP}}[V] - D_{x,\mu,a}^{\text{model}}[V]$ are an additional probe of the accuracy of each model used for parametrization. As shown in Fig. 11, the distributions narrow with reduced error going to finer lattices, with all parametrizations becoming more accurate. The L-CNN model has the sharpest distributions of all across all bare couplings, even at $\beta_{\text{wil}} = 20.0$, the range where the APE444 parametrization was optimally designed. The superiority of the L-CNN model at $\beta_{\text{wil}} = 6.0$ is particularly interesting, as this corresponds to a lattice spacing $a \sim 0.1$ fm in the range of coarsest lattice spacings used in current large-scale simulations.

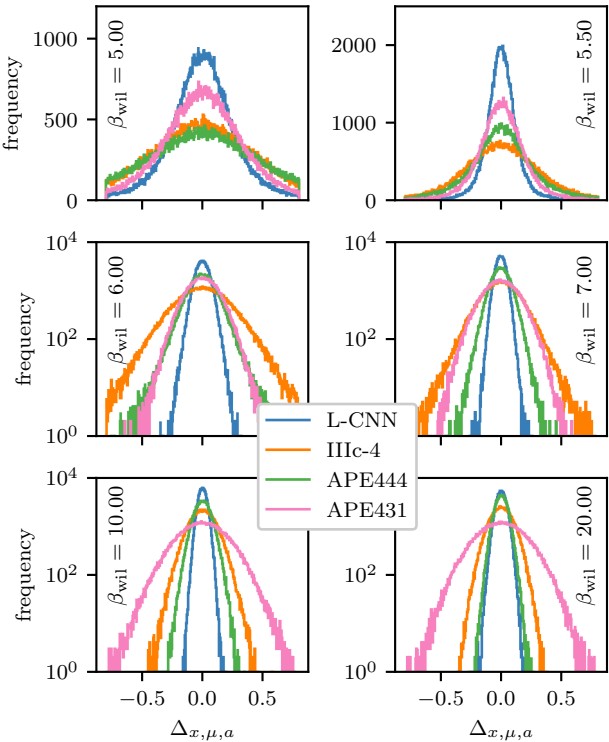

Figure 11: Histograms of the local deviations $\Delta_{x,\mu,a} = D^{\text{FP}}_{x,\mu,a} - D^{\text{model}}_{x,\mu,a}$ of the model derivative $D^{\text{model}}_{x,\mu,a}$ and the derivative of the fixed point action $D^{\text{FP}}_{x,\mu,a}$. Here, the model can refer to an L-CNN or a different parametrization of the FP action. We show the same parametrizations as in Fig. 3 for specific values of $\beta_{\text{wil}}$. The horizontal axes have been rescaled by the standard deviation of $D^{\text{FP}}_{x,\mu,a}$.

According to arguments of universality, locality of the discretized theory guarantees the correct continuum limit. While the exact FP action has infinitely many couplings, it is still a local action because the couplings decrease exponentially with the separation $r = |x - y|$ of the gauge links at positions $x$ and $y$, as shown for the perturbative couplings $\rho_{\mu\nu}(r)$ in Fig. 5. To test if the optimal L-CNN model shares this feature beyond the perturbative regime, we look at a quantity analogous to the perturbative coupling, namely the variation of the action $\mathcal{A}^{\text{L-CNN}}[V]$ with respect to gauge links at locations $x$ and $y$ and in directions $\mu$ and $\nu$. A gauge invariant observable $\hat{\rho}_{\mu\nu}(r)$ can be built from the square of this second-order derivative. The behavior of this coupling for the L-CNN model is shown in Fig. 12, measured on $6^4$ volumes at $\beta_{\text{wil}} = 6.0$ and normalized by $\hat{\rho}_{00}(0)$. The couplings do indeed decrease rapidly with separation, with a relative change of $10^{-5}$ by separation $r/a = 4$. From this, we deduce that the L-CNN network does not significantly couple fields at large separation and that the finite extent of the model does not lead to poor accuracy. We note that the numerical evaluation of the locality measure requires double-precision arithmetic to resolve the small couplings at large distances.

### C.3 RESTRICTED TRAINING RANGES

We also investigate how the selection of training data affects the performance of the L-CNN to make accurate predictions. To do so, we split the training data into *low* $\beta$ values $\beta_{\text{wil}} \in [5, 7]$ and *high* values $\beta_{\text{wil}} \in [7, 20]$, and train multiple models with random initializations on the low, high and original $\beta_{\text{wil}}$ ranges. The results are shown in Fig. 13 and Table 2. We find that each model generally performs well on the data it has been trained on. Surprisingly, the model trained on the full range performs best on coarse configurations. This might be due to the fact that this *full* model has been trained with the most data. On the other hand, the *high* model works best on high $\beta$ values. Comparing the *low* and *high* models, these results might suggest a lack of generalization of our

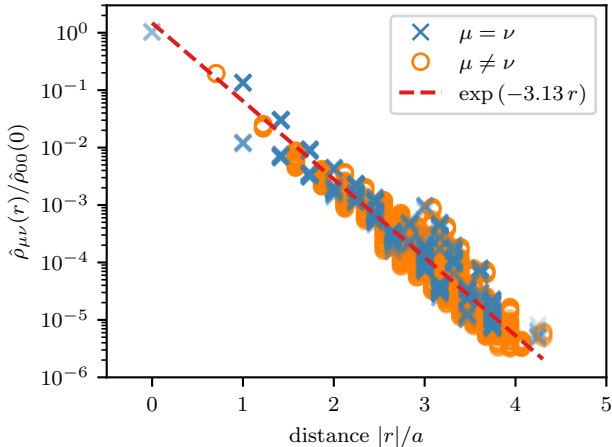

Figure 12: Locality measure $\langle \hat{\rho}_{\mu\nu}(r) \rangle$ as a function of distance $|r|$ of our best L-CNN model. The expectation value has been evaluated on five configurations at $\beta_{\mathrm{wil}} = 6.0$ on a $6^4$ lattice. Blue crosses show parallel couplings $\hat{\rho}_{\mu\mu}$, whereas orange circles correspond to orthogonal couplings $\hat{\rho}_{\mu\nu}$ with $\mu \neq \nu$. An exponential fit is shown as a red dashed line. Couplings beyond $r_{\max} \approx 4.3\,a$ are zero due to the finite receptive field of the L-CNN.

Table 2: Effect of training data selection on model performance. The left column denotes the range of $\beta_{\mathrm{wil}}$ used for evaluation, whereas the first row shows the range for training. We report the relative error of the predicted action with respect to numerical FP data, averaged over all configurations within the respective $\beta_{\mathrm{wil}}$-range. The smallest error in each column is highlighted in bold. It is apparent that the model performance strongly depends on the training range and that there is a trade-off between accuracy on particular ensembles and generality across many ensembles.

|  | test data range | | |
| --- | --- | --- | --- |
| **training data range** | $[5, 7]$ | $[7, 20]$ | $[5, 20]$ |
| $[5, 7]$ | 0.298 % | 1.787 % | 0.827 % |
| $[7, 20]$ | 2.432 % | **0.033 %** | 1.702 % |
| $[5, 20]$ | **0.217 %** | 0.138 % | **0.195 %** |

models to data outside the original training range. Models only trained on very coarse configurations tend to make less accurate predictions for smooth configurations and *vice versa*. In a sense, this suggests that despite overall good performance, the L-CNN does not truly learn the FP action that underlies the training data. However, it is unlikely that this would hinder the practical use of an FP parametrization based on L-CNNs or that this is a problem affecting only L-CNN models. Similarly, parametrizations based on simple loops as in Eq. (20) and even more sophisticated approaches using asymmetrically smeared links such as APE431 and APE444 require data from a large range of $\beta_{\mathrm{wil}}$ in order to determine suitable coefficients with good accuracy for both coarse and smooth configurations. From a practical viewpoint, especially concerning the use of FP parametrizations in a Monte Carlo simulation, it might not even be necessary to have a model that generalizes to all values of $\beta_{\mathrm{wil}}$. If one intends to perform a simulation at a particular $\beta$, it is sufficient to use a parametrization that works well on a specific level of coarseness. Much coarser and much smoother configurations are both unlikely to occur during the simulation and thus less than optimal performance outside a particular $\beta$-range does not pose a problem in practice. We also stress that the L-CNN models approach the continuum limit by construction, i.e., for sufficiently smooth fields our models reproduce the Yang-Mills action.

## C.4 FINETUNING FOR DIFFERENT LATTICE SIZES

Up until now, we have only considered models trained and tested on $4^4$ lattices. We employ transfer learning to our best type IIIc L-CNN model obtained in the last section (before finetuning on in-

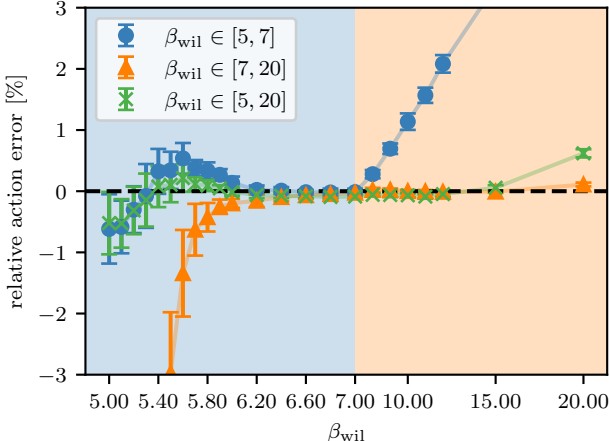

Figure 13: Effect of data selection on trained models. We show the average relative error on $4^4$ lattices of three different models for each MC ensemble from $\beta_{\text{wil}} = 5$ to $\beta_{\text{wil}} = 20$. The models have been trained on different data: low ($\beta_{\text{wil}} \in [5, 7]$, light blue region), high ($\beta_{\text{wil}} \in [7, 20]$, light orange region), and the full range ($\beta_{\text{wil}} \in [5, 20]$). The averages across all $\beta_{\text{wil}}$ are reported in Table 2.

stantons) by additional training with data from larger lattices. Specifically, we finetune on $6^4$ and $8^4$ in the range $\beta_{\text{wil}} \in [5, 20]$ for 400 epochs, starting from our previous best model. For better comparison, we also finetune our previous best model on $4^4$ with the same number of epochs. Through experimentation, we found that it is beneficial to change the loss function during this finetuning procedure. In contrast to Eq. (27), which optimizes the absolute errors of the action values and derivatives, we opt for a new loss function based on relative errors:

$$
\begin{aligned}
\mathcal{L}'_1 &= \sum_i \frac{|\mathcal{A}^{\text{FP}}[V_i] - \mathcal{A}^{\text{L-CNN}}[V_i]|}{\mathcal{A}^{\text{FP}}[V_i]}, \\
\mathcal{L}'_2 &= \sum_i \frac{\sum_{x,\mu} \text{Tr}\left[(D_{x,\mu}^{\text{FP}}[V_i] - D_{x,\mu}^{\text{L-CNN}}[V_i])^2\right]}{\sum_{x,\mu} \text{Tr}\left[(D_{x,\mu}^{\text{FP}}[V_i])^2\right]}, \\
\mathcal{L}' &= w'_1 \mathcal{L}'_1 + w'_2 \mathcal{L}'_2,
\end{aligned}
\tag{28}
$$

with typical choices $w'_1 = w'_2 = 1$.

The performance of these three finetuned models can then be compared across different lattice sizes. The results are summarized in Table 3, where we list the relative error measured by the action values and the gauge invariant derivative error for each lattice size. Remarkably, we find that the performance improves only slightly with additional transfer learning and is consistent for all considered lattice sizes. This suggests that training on small lattice sizes is sufficient to obtain FP parametrizations with high accuracy, which generalize beyond the original training data in terms of lattice size. This is highly advantageous because training on small lattices is much more efficient: a typical model trained for 100 epochs requires approximately 4 hours on $4^4$, 7 hours on $6^4$, and 22 hours on $8^4$ on an NVidia 3090 RTX GPU.

## C.5    FINETUNING WITH INSTANTONS

We have seen that the performance of a trained model strongly depends on the properties of the training configurations. The largest effect stems from the coarseness of equilibrated configurations, controlled by $\beta_{\text{wil}}$, as demonstrated in Sec. C.3. We may extend our training procedure to also include nonequilibrium configurations, for example instanton solutions. One might expect that an L-CNN trained to sub-percent accuracy within $\beta_{\text{wil}} \in [5, 20]$ would also produce similarly accurate predictions for instantons, but we find that this is not necessarily the case. If instantons are absent during training, then predictions for their FP action values appear to be mostly determined by the

Table 3: Effect of transfer learning with different lattice sizes. Starting from our previous best model, we use transfer learning to obtain models that have been finetuned to $4^4$, $6^4$, and $8^4$ data. The left column denotes three different models and we report the relative error and derivative error on various lattice sizes for $\beta_{\mathrm{wil}} \in [5, 20]$. The smallest errors in each column are highlighted in bold. The lattice size appears to have a negligible effect on model performance.

| | relative error (test data) | | |
|---|---|---|---|
| **finetuned model** | $4^4$ | $6^4$ | $8^4$ |
| $4^4$ | **0.178 %** | 0.201 % | 0.181 % |
| $6^4$ | 0.185 % | **0.196 %** | 0.177 % |
| $8^4$ | 0.191 % | 0.202 % | **0.176 %** |
| | derivative error (test data) | | |
| **finetuned model** | $4^4$ | $6^4$ | $8^4$ |
| $4^4$ | $\mathbf{7.63 \times 10^{-2}}$ | $8.19 \times 10^{-2}$ | $8.22 \times 10^{-2}$ |
| $6^4$ | $\mathbf{7.39 \times 10^{-2}}$ | $7.93 \times 10^{-2}$ | $7.96 \times 10^{-2}$ |
| $8^4$ | $\mathbf{7.36 \times 10^{-2}}$ | $7.91 \times 10^{-2}$ | $7.93 \times 10^{-2}$ |

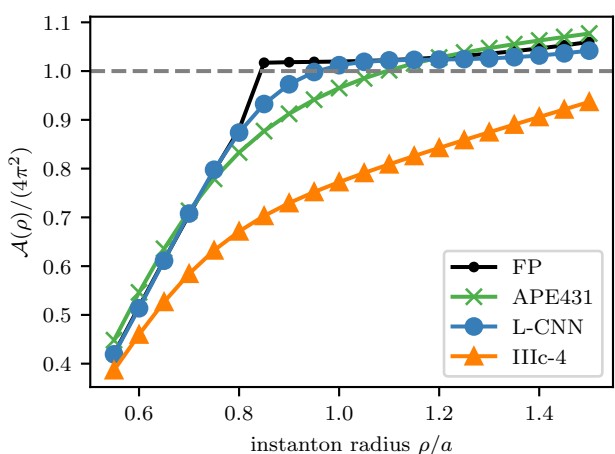

Figure 14: Evaluating different parametrizations of the FP action on instanton configurations with radii $\rho/a$ on an $8^4$ lattice. The black points show the numerical fixed point data. The L-CNN model is a type IIIc-inspired action with Symanzik-constrained trainable parameters. As detailed in the main text, it has been finetuned on instanton configurations.

prefactor action $\mathcal{A}^{\mathrm{pre}}[V]$. In the case of the IIIc-4 prefactor action, we find a relative error of $\sim 10\%$ for instanton radii between $\rho/a = 0.5$ and $\rho/a = 1.5$.

Predictions for instantons can be drastically improved by including them as training configurations in the finetuning procedure. Starting from our best $4^4$ model found in Sec. C.4, we extend the training data set from equilibrated configurations within $\beta_{\mathrm{wil}} \in [5, 20]$ to include 20 different instantons and perform transfer learning with a reduced learning rate and increased batch size for 1000 additional epochs with $w_1' = 1$ and $w_2' = 0.1$. This parameter choice puts more weight on accurate action values at the cost of slightly more inaccurate predictions for derivatives. To avoid data imbalance, the instantons are included multiple times such that we obtain effectively 200 training instantons.

We test our finetuned model on instantons of various sizes. The results are shown in Fig. 14, where we plot the predicted action as a function of the instanton radius. We see that our model predicts the numerical FP data much better than the IIIc-4 action and even the APE431 action. The predictions closely follow the FP data, except for the kink around $\rho/a = 0.85$. Moreover, we find that our finetuning procedure does not lead to a loss of performance on equilibrated configurations. Our finetuned model has a relative error of $0.12\%$ and a gauge-invariant derivative error $\mathcal{L}_2 = 8.731 \cdot 10^{-2}$ within $\beta_{\mathrm{wil}} \in [5, 20]$. We note that this finetuned model is the one presented in Sec. C.2.

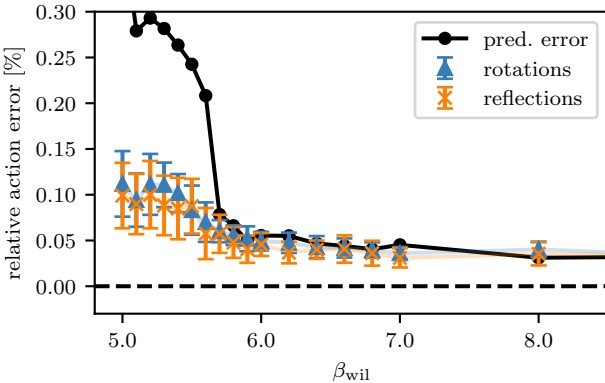

Figure 15: Relative error due to breaking of rotational and reflection symmetry as a function of $\beta_{\text{wil}}$. We also show the relative prediction error for comparison (black dots).

### C.6 APPROXIMATE LATTICE SYMMETRIES

Finally, we may check the trained model for discrete lattice symmetries. The L-CNN used in this work is, by construction, equivariant with respect to lattice translations. As a result, if $U$ and $U'_{(\text{shift})}$ are two gauge configurations which are the same up to a shift on the lattice, then the predictions will agree exactly

$$\mathcal{A}^{\text{L-CNN}}[U'_{(\text{shift})}] = \mathcal{A}^{\text{L-CNN}}[U]. \tag{29}$$

On the other hand, other lattice symmetries such as rotations and reflections are not implemented exactly. A rotated gauge configuration $U'_{(\text{rot})}$ is generally assigned a different action value

$$\mathcal{A}^{\text{L-CNN}}[U'_{(\text{rot})}] \neq \mathcal{A}^{\text{L-CNN}}[U]. \tag{30}$$

In principle, the L-CNN architecture can be extended to include such discrete lattice symmetries exactly (Aronsson et al., 2023), but only at considerable computational cost. Thus, with the goal in mind to use the trained model in a future Monte Carlo study, we only consider the more efficient translationally-equivariant L-CNNs and test symmetry properties after training.

For rotational invariance, we consider all $90°$ rotations about a single origin on the lattice. Taking into account both clockwise and counter-clockwise rotations, these amount to $D(D-1)$ transformations in $D$ lattice dimensions. The choice of origin is arbitrary due to translational equivariance. Given a particular gauge configuration $U_{(0)}$ from the test set, we generate the set of rotated configurations $U_{(j)}$ with $j \in \{1, 2, \ldots, D(D-1)\}$. For each of these configurations, we compute the predicted action $\mathcal{A}_{(j)} = \mathcal{A}^{\text{L-CNN}}[U_{(j)}]$. We then define the relative error due to broken rotational invariance as the standard deviation of the set $\{\mathcal{A}_{(j)}\}$ normalized to the mean value. A similar measure can be defined for reflections along lattice axes.

We present our results in Fig. 15, where we evaluate the measures for broken symmetry on equilibrated configurations on a $4^4$ lattice from $\beta_{\text{wil}} = 5.0$ to $8.0$. We observe that the variance between predictions due to symmetry transformations (either rotations or reflections) is much smaller than the prediction error for coarse configurations ($\beta_{\text{wil}} < 6$). For smoother configurations, the errors become comparable. Overall, we conclude that sufficiently well-trained models exhibit approximate rotation and reflection symmetry. These symmetries are *a priori* not present in the L-CNN architecture and have been learned during training.

## D  AUTOMATIC GROUP DIFFERENTIATION

Training parametrized actions, i.e., minimizing the loss function in Eq. (27), requires efficient methods to compute exact group derivatives of actions as defined in Eq. (11). In this appendix, we show

how group derivatives are related to Wirtinger derivatives, which can be computed using backpropagation.

Given a complex scalar function $f : \mathbb{C} \to \mathbb{C}$, the Wirtinger derivatives are defined by

$$\frac{\partial f}{\partial z} = \frac{1}{2}\left(\frac{\partial f}{\partial x} - i\frac{\partial f}{\partial y}\right), \quad \frac{\partial f}{\partial \bar{z}} = \frac{1}{2}\left(\frac{\partial f}{\partial x} + i\frac{\partial f}{\partial y}\right),\tag{31}$$

where $z = x + iy$, $x, y \in \mathbb{R}$ and $\bar{z}$ is the complex conjugate. Extending this definition to scalar functions of complex matrices $U$ we use

$$\left(\frac{\partial f}{\partial U}\right)_{ij} = \frac{1}{2}\left(\frac{\partial f}{\partial \mathrm{Re}(U)_{ji}} - i\frac{\partial f}{\partial \mathrm{Im}(U)_{ji}}\right),\tag{32}$$

$$\left(\frac{\partial f}{\partial U^\dagger}\right)_{ij} = \frac{1}{2}\left(\frac{\partial f}{\partial \mathrm{Re}(U)_{ij}} + i\frac{\partial f}{\partial \mathrm{Im}(U)_{ij}}\right).\tag{33}$$

Using these definitions, we obtain a compact expression for the Taylor expansion of $f$ around $U' = U + \epsilon\delta U$ up to linear order in $\epsilon \ll 1$:

$$f(U + \epsilon\delta U) = f(U) + \epsilon\,\mathrm{Tr}\left[\frac{\partial f}{\partial U}\delta U + \frac{\partial f}{\partial U^\dagger}\delta U^\dagger\right] + O(\epsilon^2).\tag{34}$$

In the context of functions on $\mathrm{SU}(N_c)$, the group derivative is understood as varying the matrix $U$ *along* the group manifold via

$$\frac{\delta}{\delta U^a}f(U) \equiv \lim_{\epsilon \to 0}\frac{1}{\epsilon}\left(f(e^{i\epsilon t^a}U) - f(U)\right)$$
$$= \frac{d}{d\epsilon}f(e^{i\epsilon t^a}U)\big|_{\epsilon=0}.\tag{35}$$

By expanding the matrix exponential in $\epsilon$, we find that this corresponds to a variation matrix $\delta U = it^a U$. Inserting this into Eq. (34), we find

$$f(e^{i\epsilon t^a}U) \approx f(U + i\epsilon t^a U)\tag{36}$$

$$= f(U) + i\epsilon\,\mathrm{Tr}\left[\left(U\frac{\partial f}{\partial U} - \frac{\partial f}{\partial U^\dagger}U^\dagger\right)t^a\right] + O(\epsilon^2).\tag{37}$$

Thus, Eq. (35) becomes

$$\frac{\delta}{\delta U^a}f(U) = i\,\mathrm{Tr}\left[\left(U\frac{\partial f}{\partial U} - \frac{\partial f}{\partial U^\dagger}U^\dagger\right)t^a\right].\tag{38}$$

A relevant example is the function

$$f(U) = \mathrm{ReTr}[UW] = \frac{1}{2}\left(\mathrm{Tr}[UW] + \mathrm{Tr}[W^\dagger U^\dagger]\right),\tag{39}$$

for which the Wirtinger matrix derivatives are

$$\frac{\partial f}{\partial U} = \frac{1}{2}W, \quad \frac{\partial f}{\partial U^\dagger} = \frac{1}{2}W^\dagger.\tag{40}$$

Insertion into Eq. (38) yields

$$\frac{\delta}{\delta U^a}f(U) = \frac{i}{2}\,\mathrm{Tr}\left[\left(UW - W^\dagger U^\dagger\right)t^a\right] = \mathrm{ReTr}\left[iUWt^a\right]\tag{41}$$

analogous to the FP action derivative and its connection to the blocking kernel as in Eq. (10). This result enables the use of exact group derivatives of parametrized actions, because the automatic differentiation engine of *PyTorch* is able to compute matrix-valued Wirtinger derivatives of arbitrary differentiable functions.

