# OpenReview forum: "Application of gauge equivariant convolutional neural networks to learning a fixed point action for SU(3) gauge theory"
_ICLR.cc/2024/Workshop/AI4DiffEqtnsInSci — AI4DiffEqtnsInSci @ ICLR 2024 Poster_

### Official Review · Reviewer_eZaP · 2024-02-25
**Review of Application of gauge equivariant convolutional neural networks to learning a fixed point action for SU(3) gauge theory**

**Rating:** 7
**Confidence:** 2

**Review:**

The paper presents an innovative use of Gauge Equivariant Convolutional Neural Networks (L-CNNs) for determining Fixed Point actions within SU(3) gauge theory, addressing critical computational challenges in nuclear physics simulations. This method not only preserves gauge symmetry but also showcases a more efficient parametrization than traditional approaches, marking a significant step forward in the field.

This work provides a novel application of Gauge Equivariant Convolutional Neural Networks (L-CNN), demonstrating a more efficient parametrization of Fixed Point actions. The paper provides an in depth investigation of L-CNNs for the chosen task and, while the results expectedly show that the bigger the network the better it works, it is an appreciated exploration.

It would have been interesting to also have a comment on the generalizability of the model, together with the challenges of implementing the model for such a specific task and how would this hinder future explorations.

Despite these areas for potential expansion, the paper is well-composed and the research meticulously conducted. Its contributions to computational physics are clear, and it stands to inspire further innovation. I recommend acceptance.

---

### Meta-Review · Area_Chair_chyx · 2024-03-03

**Recommendation:** Accept (Poster)

**Metareview:**

The reviewer marks this paper as a clear acceptance. I also vote for acceptance and recommend that the authors address the reviewer's comments in the camera-ready version.

---

### Decision · Program_Chairs · 2024-03-03

Accept (Poster)